# Amount and type of physical activity and sports from one year forward after hip or knee arthroplasty—A systematic review

Yvet Mooiweer, Inge van den Akker-Scheek, Martin Stevens *, On behalf of the PAIR study group¶

Department of Orthopedics, University of Groningen, University Medical Center Groningen, Groningen, The Netherlands

¶ Membership of the PAIR study group is listed in the Acknowledgments.
* m.stevens@umcg.nl

## Abstract

### Introduction

After rehabilitation following total hip or knee arthroplasty (THA/TKA), patients are advised to participate in physical activity (PA) and sports. However, profound insight into whether people adopt a physically active lifestyle is lacking. Aim is to gain insight into the performed amount and type of PA (including sports) and time spent sedentarily by persons after THA/TKA.

### Methods

A systematic review (PROSPERO: CRD42020178556). Pubmed, Cinahl, EMBASE and PsycInfo were systematically searched for articles reporting on amount of PA, and on the kind of activities performed between January 1995-January 2021. Quality of the articles was assessed with the adapted tool from Borghouts et al.

### Results

The search retrieved 5029 articles, leading to inclusion of 125 articles reporting data of 123 groups; 53 articles reported on subjects post-THA, 16 on post-hip-resurfacing arthroplasty, 40 on post-TKA, 15 on post-unicompartimental knee arthroplasty and 12 on a mix of arthroplasty types. With respect to quality assessment, 14 articles (11%) met three or fewer criteria, 29 (24%) met four, 32 (26%) met five, 42 (34%) met six, and 6 (5%) met seven out of the eight criteria. PA levels were comparable for THA and TKA, showing a low to moderately active population. Time spent was mostly of low intensity. Roughly 50% of -subjects met health-enhancing PA guidelines. They spent the largest part of their day sedentarily. Sports participation was relatively high (rates above 70%). Most participation was in low-impact sports at a recreational level. Roughly speaking, participants were engaged in sports 3 hours/week, consisting of about three 1-hour sessions.

**Data Availability Statement:** All relevant data are within the manuscript and its Supporting Information files.

**Funding:** This project has been funded by the Erasmus+ programme of the European Commission (613008-EPP-1-2019-IT-SPO-SCP). The funders had no role in study design, data collection and analysis, decision to publish, or preparation of the manuscript.

**Competing interests:** The authors have declared that no competing interests exist.

## Conclusion

Activity levels seem to be low; less than half of them seemed to perform the advised amount of PA following health-enhancing guidelines Sports participation levels were high. However, many articles were unclear about the definition of sports participation, which could have led to overestimation.

## Introduction

Total hip arthroplasty (THA) and total knee arthroplasty (TKA) are cost-effective and pain-relieving treatments for end-stage osteoarthritis, and improve the ability to stay physically active [1, 2]. After THA and TKA sufficient participation in physical activity (PA), including sports, is of importance, not only from a general health perspective but also because PA benefits the functioning and motor control of the prosthetic joint [3–5]. A physically active lifestyle induces commonly known physical and mental health benefits such as lower risk for several non-communicable diseases and improved cognitive health [6–9]. Being physically active also improves physical fitness, which is necessary to perform activities of daily living – of major importance in the older age group on which most THAs and TKAs are performed. PA has additional benefits for persons after THA or TKA as it leads to improved fixation of the implant, improved bone density and a lower fall risk [3–5]. On the other hand, a physically inactive or sedentary lifestyle be conducive to additional health problems. Besides, it can lead to overweight or even obesity, and in turn possibly to increased wear of the prosthesis due to the greater load.

Recognition of the importance of PA has led to international recommendations for health-enhancing PA. Until 1995, recommendations focused primarily on the development and maintenance of cardiorespiratory and muscular fitness. In 1995 a paradigm shift occurred. The American College of Sports Medicine (ACSM), together with the Centers for Disease Control (CDC) in the United States, released new recommendations [10]. These new recommendations are characterized by their primary focus on the relationship between PA and health-related benefits. Nowadays, health-enhancing PA recommendations have evolved into the most recent recommendations of the World Health Organization, recommending 150–300 minutes of moderate-intensity PA/week or 75–150 minutes of vigorous PA/week, or an equivalent combination. Additionally, twice a week bone- and muscle-strengthening exercises should be performed, which for older adults should be combined with balance exercises. Finally, sedentary behavior should be reduced [11]. Despite these recommendations, studies show that a large part of the European population still does not meet these recommendations [12]. Information is scarce on whether persons after THA and TKA are physically active and meet these recommendations.

In the past, reviews have been published about PA after THA or TKA, but they focused on the difference between pre-surgery and post-surgery [13–15]. Those reviews reveal that the difference in the amount of PA performed does not increase or only slightly increases until one year after THA or TKA [16–19]. Information about whether a physically active lifestyle is adopted after the replacement of a painful hip or knee joint is lacking in these systematic reviews. Aim of this systematic review is therefore to gain insight into the performed amount and type of PA (including sports) and the time spent sedentarily by persons from one year forward after THA or TKA.

## Methods

### Search strategy

A systematic review with a narrative synthesis was conducted. The review was registered in Prospero (PROSPERO: CRD42020178556) beforehand. A librarian of the Central Medical Library of University Medical Center Groningen (UMCG) was consulted for the search strategy. The final search strategy is shown in S1 Appendix.

### Study selection

The PubMed, Embase, CINAHL and PsycInfo databases were systematically searched for 1) articles (excluding review articles, case reports and study protocols) reporting on amount of PA (including sport activities) performed by subjects after completing rehabilitation for THA or TKA, and 2) articles reporting on the kind of physical or sport activities performed by subjects after completion of rehabilitation for THA or TKA. Included subjects had to be over 18 years of age at the time of the measurements. Articles written in a language other than English, review articles, case reports and study protocols were excluded. For the question about the amount of PA performed, an additional criterion was that participants were not allowed to have been involved in an intervention program specifically designed to influence their PA behavior. The initial search was performed on 26 March 2020 and updated on 13 January 2021. Articles were searched back to 1 January 2010. One year after surgery was taken as the endpoint of the rehabilitation period, as studies show that individual's PA still improves between 6 and 12 months postoperatively [16–19].

Articles identified by the search strategy were imported to EndNote X9 (Clarivate Analytics Endnote X9.3.1, Philadelphia) and duplicates were removed following the guidelines proposed by Bramer et al. 2016 [20]. Articles were first screened for eligibility based on title and abstract. All articles selected by the authors were screened for eligibility through full-text reading. The screening procedure was performed by two authors (Y.M. and R.G.) independently, differences were solved by discussion, and when needed a third assessor (M.S.) was consulted.

### Data extraction and analysis

Data extraction was performed and data was included in several data extraction tables. First, a general table was created which included information about author and year, country, participant characteristics, measurements used, outcomes and results. After creation of this general table including all information extracted from all articles, four separate tables were created out of the general table, each covering a certain aspect of PA or sport information. The categories of the four tables were "amount of PA", "amount of activity measured by scales", "amount of sport participation" and "sport type". Data was sorted first by arthroplasty type, then by measurement tool. Data extraction was performed by the first author (Y.M.), and when deemed necessary because of uncertainty about the interpretation of the data a second assessor was involved (R.G.).

The methodological quality of the included studies was independently assessed by two authors (Y.M. & R.G.). In case of discrepancies, these were solved after discussion, and if needed after consultation of a third reviewer (M.S.). To assess the methodological quality of the studies, a tool adapted from the tool used by Borghouts et al. 1998 [21] was used. The adaptation aimed at focusing on the representativeness of the included population and the validity of the outcomes. The final tool consists of a total of 8 questions, each worth 1 point. The final tool is shown in S2 Appendix.

# Results

The search strategy retrieved 7759 articles of which 2730 were duplicates, so 5029 articles remained for title and abstract screening. Of these articles, 146 were selected for full-text screening, ending in inclusion of 125 articles [3, 4, 16, 22–142] reporting data of 123 unique groups. More details about the inclusion can be found in the flowchart in Fig 1. Of the included articles, 53 reported on subjects post-THA, 16 on post-hip-resurfacing arthroplasty (HRA), 40 on post-TKA, 15 on post-unicompartimental knee arthroplasty (UKA), and 12 on a mix of arthroplasty types without separating the results.

## Methodological quality

Results of the quality assessment of the included studies can be found in S3 Appendix. The 125 articles included 123 different studies, of which 14 (11%) met three or less of the criteria, 29 (24%) met four criteria, 32 (26%) met five criteria, 42 (34%) met six criteria and 6 (5%) articles met seven criteria. None of the studies met all eight criteria. Methodological quality was no reason to exclude articles from the analysis.

Except for thirteen (11%) studies, inclusion and exclusion criteria were clearly defined. Eighty-seven (71%) of the studies included participants consecutively or at random, with 27 (22%) lacking clarity about their inclusion method and nine (7%) using another mode of inclusion that was not random or consecutive. Response rates differed: 47 (38%) studies had a sufficient response rate compared to 53 (43%) with a non-sufficient response rate, whereas 23 (19%) studies lacked clarity as to how many of the potentially eligible subjects did not respond. Thirty-three (27%) studies performed a non-response analysis, while 74 (60%) did not; the remaining 16 (13%) studies included all participants invited and thus did not need to perform a response analysis. Eventually, the study sizes were sufficient in 65 (53%) studies, compared to 58 (47%) studies with a smaller study size. Most of the studies used only questionnaires (98, (80%), with 25 (20%) studies using objective measurements. Quality of the outcome measures used was sufficiently in 115 (93%) studies, and reporting of the outcome measures was sufficient in 108 (88%) of the 123 studies.

## Amount and type of physical activity

All results extracted from the articles can be found in S4 Appendix. Amount of PA was measured in three different ways. First, with the help of objective measurement methods like accelerometers, pedometers, or comparable technical devices time spent on PA and sometimes intensity was measured. Second, patient-reported questionnaires were used requesting participants to report the time they spent on PA and sometimes intensity of activities. S5 Appendix shows the results on amount and intensity of PA. Third, one-item questionnaires were used that give a general indication of the overall PA performed; these usually are a combination of both time and intensity, with a higher score representing more (and more intense) PA. These results can be found in S6 Appendix.

Results considering sports activity were categorized into levels of sports participation, time spent playing sports (including session length and frequency), and the impact and intensity of the sports, and can be found in S7 Appendix. The participation rates per individual sport can be found in S8 Appendix.

## Physical activity after total hip arthroplasty

Number of steps taken was assessed in 11 articles in 10 unique groups. All articles used objective measurement methods, two in combination with a questionnaire [43, 62, 63, 75, 87, 114,

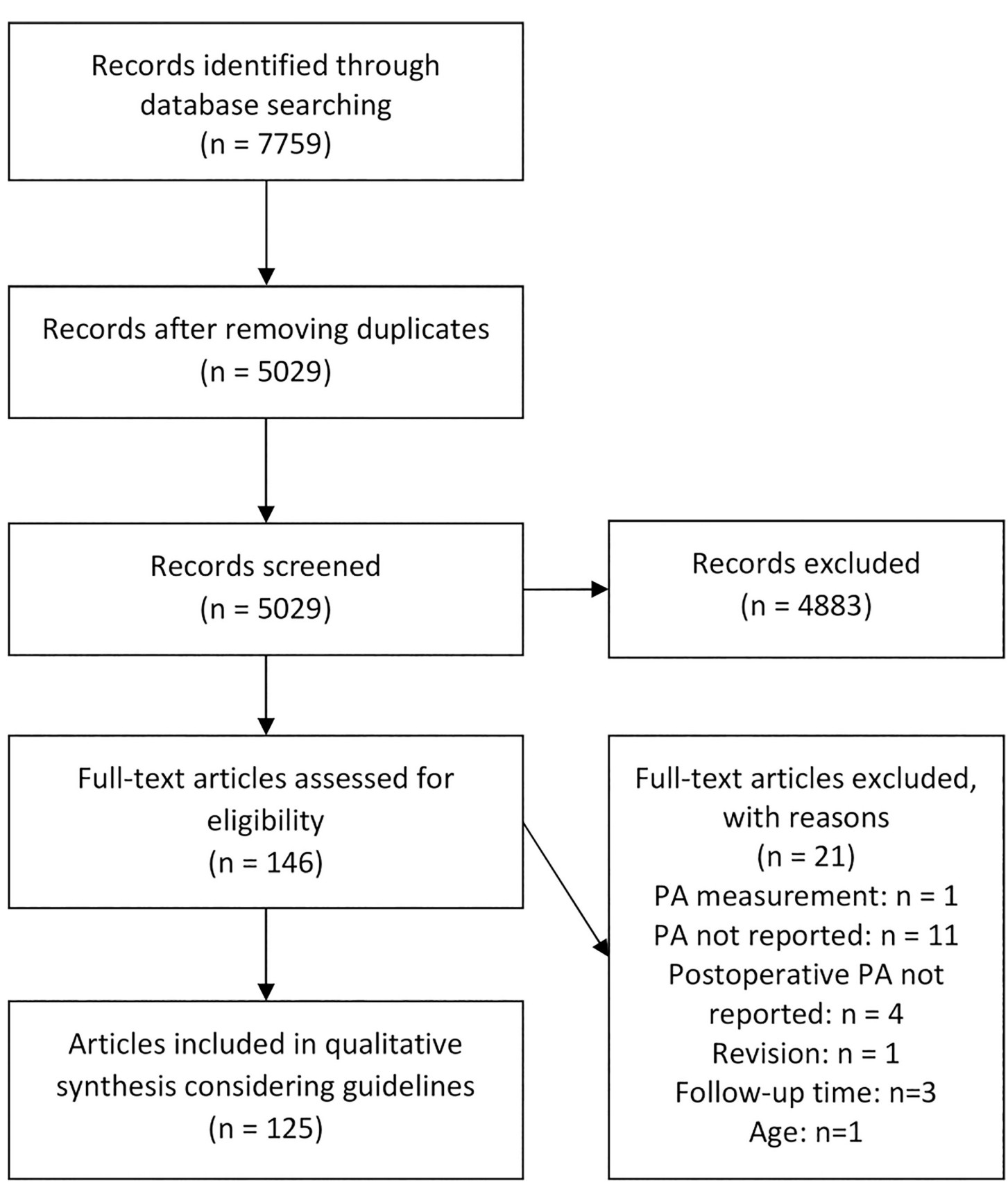

**Fig 1. Flow chart of study inclusion.**

117, 123, 128, 132, 133]. Results showed six articles reporting an average 5000–7000 steps/day and four articles (three studies) reporting means between 4000 and 5000 steps/day. Clement et al. [128] reported the steps/day in three different age groups at 12 and 24 months postoperatively. Subjects under age 65 at 12 months postoperatively and those between ages 65 and 74 at 12 as well as 24 months postoperatively had 5000–7000 steps/day, subjects under age 65 at 24 months postoperatively had 4000–5000 steps/day, and subjects 75 or older had a mean of 3915 steps/day.

Six articles reported time spent on PA during a week, including one accelerometer, one pedometer and three questionnaire studies [23, 38, 43, 96, 106, 118]. A high range of outcomes were found, from Wagenmakers et al. [118] reporting 1468.1 ± 1138.3 minutes of activity per week using the SQUASH questionnaire to Alvarez et al. [23] reporting 148.9 ± 69.8 min/week of activity using an accelerometer. Health-enhancing PA guidelines were met by 18% of the subjects of Matsunaga-Myoji et al. [19] using an accelerometer and by 50% of the subjects of Paxton et al. [96] through direct questioning by the nursing staff, while Wagenmakers et al. [118] reported 67% meeting guidelines using the SQUASH questionnaire [87, 96, 118]. Ninomiya et al. [92] classified 34.4% as highly active, based on spending 1000 or more kcal/week as assessed by the International Physical Activity Questionnaire (IPAQ). Jelsma et al. [132] classified 19% as "somewhat active" based on 8000 steps/day, while they classified 50% as sedentary based on <5000 steps/day. On average, their subjects spent 9.6 hours of wake time sitting or lying down. Clement et al. [128] also assessed the time spent sedentarily, and found for all age groups at both time points that participants sat on average 17–19 hours/day.

Three articles reporting on how much activity was performed at which intensity all showed that low-intensity physical activity (LPA) formed the largest part of the total PA performed, at 72%, and twice 55% for LPA [38, 43, 118]. Further, Kuhn et al. [75] reported participants to be inactive 71.1% of the time, performing LPA 18.7% of the time, and spending the remaining 10.2% on moderate-to-vigorous physical activity (MVPA). A significant increase from 58.3 ± 64.6 to 72.3 ± 67.4 minutes of MVPA per week was found 1–3 years postoperatively by Matsunaga-Myoji et al. [87].

Thirty-one articles reported on activity using one-item questionnaires that give a general indication of activity [23, 26, 36, 37, 41, 44, 47–49, 51, 57, 61, 64, 67, 71, 75, 85, 90, 93, 94, 103, 107, 110, 114, 115, 123, 126, 128, 136, 138, 139]. On the University of California Los Angeles activity score (UCLA) questionnaire, 21 articles reported a score between 5.5 and 7, which means regular participation in moderate activities. Four articles had a mean UCLA score below 5.5 points and five articles above 7.5 points. Nine articles used similar scores (Tegner activity scale, Grimby scale, Lower Extremity Activity Scale (LEAS), Sports Activity Index, Weighted Activity Score) and their results are comparable to those of the studies using the UCLA [34, 36, 44, 56, 84, 100, 114].

## Sports after total hip arthroplasty

Sports participation was reported in eleven articles: in nine participation rates varied between 64% to 91% and two articles reported lower percentages [29, 37, 47, 57, 83, 93, 94, 110, 125, 126, 136]. Hara et al. [47] reported that 30.5% participated in sports, while Madrid et al. [83] reported 7.1% institutionalized sport participation. Six of the articles reporting on sports participation did not define sports participation in terms of frequency and intensity. Three other articles [29, 37, 125] required regular sports participation for study inclusion, while Hara et al. [47] had a once-a-month inclusion requirement. Madrid et al. [83] did have an inclusion requirement to participate in institutionalized sports but did not elaborate further on frequency of participation. Participation in sports was largely recreational, with Ortmaier et al.

[94] reporting that more than 80% of the sports were performed recreationally and Bonnin et al. [29] reporting that only 2.6% participated in competitive sports. Lefevre et al. [77] reported that all judokas stopped competitive judo after undergoing THA or TKA.

Mean time spent playing sports was only reported by Donner et al. [37], at 4.2 hours/week divided over 3.4 ± 2.9 times/week. Using questionnaires, two articles reported on time spent playing sports [18, 83]: Madrid et al. [83] reported 42% subjects playing <5 hours/week. Smith et al. [18] reported 2–4 hours/week for 67% of subjects participating in low-intensity sports, >2 hours/week for most subjects (78.7%) performing moderate-intensity sports, and <2 hours/week for 84.2% of subjects performing high-intensity sports. Four articles reported on mean number of sessions per week, finding 3.4 ± 2.9, 3 ± 1.0, 2.6 and 1.8 ± 1.1 [37, 51, 57, 110], with mean session lengths of about one hour [57, 110].

Type of sport was reported in twelve articles [29, 37, 47, 57, 83, 93, 94, 103, 106, 110, 125, 136]. Low-impact sports were commonly performed, with cycling, fitness, golf, gymnastics, swimming and walking reported the most. For medium-impact sports, hiking was reported by over 40% of participants in five out of nine articles [29, 93, 94, 110, 125], while other medium-impact sports were generally performed by <20% of participants. High-impact sports were performed by <10% of participants, yet Oliver et al. reported over 20% of participants engaging in ball sports and jogging, and 45% in tennis or squash.

## Physical activity after hip-resurfacing arthroplasty

Number of steps was reported in two articles measuring the same group using an accelerometer [132, 133], with a daily steps median of 5546 and range 2274 to 9966. Time spent in PA was reported by the same studies as number of steps [132, 133]. Active time was 10.8% of the accelerometer wear time. This active time was divided over a median of 1.3 hours/day spent walking and 0.05 hours/day spent cycling. Further, 3.0 hours/day was spent standing and 7.6 sitting. Results of the SQUASH questionnaire revealed a median spent of 6150 MET (metabolic equivalent of task) per week.

One-item questionnaires giving a general indication of PA were used in twelve articles [4, 25, 40, 42, 45, 64, 72, 74, 76, 86, 109, 139]. Eleven used the UCLA score. A score between 7 and 8 was most common and was reported by seven articles, referring to regular participation in active events. One article reported a lower score of 6.8 and three had a score above 9; those articles only included subjects practicing high-impact sports preoperatively [40, 42, 45, 139].

## Sports after hip-resurfacing arthroplasty

Sports participation was reported by six articles: five reported rates between 90% and 98%, and Fisher et al. reported 73% participation in sports at least once a month [25, 40, 42, 45, 76, 109]. Besides total sports participation of 97%, Banerjee et al. [25] reported 61% daily participation in sports. In the other articles sports participation was not defined in terms of intensity and frequency. Intensity of sports activity performed was reported only by Le Duff & Amstutz [76], who found that of the sports performed 5.5% were at a competitive level 1.8 years postoperatively, increasing to 10.1% at 9.1 years postoperatively.

Time spent on sports was reported by two articles, both finding on average 3 hours/week of participation [42, 45]. Two other articles reported on session length and frequency. Sandiford et al. [109] reported that all of their preoperatively active subjects participated in sports more than three times per week, with session lengths of 60–90 minutes. Of the subjects in the consecutive series of Le Duff & Amstutz [17], one third reported participation more than 12 times per month, with session lengths of 30–60 minutes.

Type of sports was studied in five articles, with Girard et al. only reporting on high-impact sports [24, 25, 40, 45, 109]. The highest participation rates were seen in low-intensity sports, especially swimming, cycling and fitness, with hiking the exception as intermediate impact sport. Participation rates in intermediate- and high-impact sports generally did not exceed 10% per discipline in most articles. Girard et al. and Sandiford et al. included highly active subjects preoperatively [45, 109], reporting higher participation rates compared to 30% and 76% in the other articles for jogging. However, Girard et al. [45] stated that all those participating in high-impact sports postoperatively had to modify their participation.

## Physical activity after total knee arthroplasty

Number of steps taken per day was measured objectively in nine articles, with consistent reports of 5900–6800 steps on average, except for Matsunaga-Myoji et al. [134] reporting 4587 steps/day and Wimmer et al. [122] reporting an average of 3102 ± 1553 steps per 12 hours [30, 35, 81, 82, 121, 122, 124, 127, 134].

Time spent actively was reported in six articles. Matsunaga-Myoji et al. [134] used an accelerometer and reported a mean PA of 372 min/week. Two studies used the SQUASH questionnaire to determine PA and found a total activity time of 1347 ± 1278 min/week and 1337 ± 1260 min/week [46, 69]. Using questioning by nursing staff, Paxton et al. [96] found that participants were active for 150 [60–280] min/week. Two articles reported average MET hours/week using questionnaires. Jones et al. [65] used the Historical Leisure Activity Questionnaire (HLAQ), finding a mean total activity of 21.4 MET hours/week. Ristolainen et al. [104] compared two groups having arthroplasty because of previous trauma, one due to sports injuries and the other due to non-sport injuries, and found a total activity of 42.1 and 18.5 MET hours/week, respectively. Eight articles reported on number of subjects meeting PA guidelines: four studies [46, 55, 69, 96] yielded around 50%, while according to accelerometer data of Lutzner et al. [82] and Bin sheeha et al. [124] about 20% met health-enhancing PA guidelines. Additionally, in the accelerometer study of Hylkema et al. [131] 70% participated in 150 minutes of MVPA or more, while 41.4% of the subjects of Matsunaga-Myoji et al. [134] met PA guidelines for older persons (MVPA $\geq$ 52.5 min/week).

Intensity of activities was reported by seven articles. In two different articles by Lutzner et al. [81, 82] about 1900 steps, which is about 30% of the total amount of steps, were reported to be of moderate-to-vigorous intensity. Two other accelerometer studies reported MVPAs of 42.8 and 41.7 min/week [124, 134], while Hylkema et al. [131] reported subjects performing LPA 36.8% and MVPA 3.1% of the wear time. Kersten et al. [69], using the SQUASH questionnaire, found that most of the activities were of low intensity, with 780 ± 874 minutes spent on light PA, 337 ± 577 min/week on moderate PA, and 223 ± 374 min/week on vigorous PA. Using the HLAQ, Jones et al. [65] reported that 19.6 MET hours/week of the total activity of 21.4 MET hours/week was due to medium-intensity activity. Information about sedentary behavior was reported by six articles [55, 81, 121, 122, 124, 131]: Bin sheeha et al. [124] reported a sedentary time of 19.1 hours per 24 hours, including lying down and thus sleeping. Webber et al. [121] reported a sedentary time of 9.2 ± 1.4 hours/day or 63.8% ± 10.0 of wear time, Wimmer et al. [122] found an average 59.9% sitting time for 12 hours of wear time, and Hylkema et al. [131] found an average time of 60.1% of wear time, all using activity monitors or accelerometers. Lutzner et al. [81] defined 34.8% of their subjects as sedentary based on setting lower than 5000 steps/day, while Hodges et al. [55] reported 45% sitting six hours or longer.

Sixteen articles reported on PA using one-item questionnaires that give a general indication of activity performed; 10 of those used the UCLA scale. Results in those articles varied, with

four reporting scores between 4.5 and 5, six between 5 and 6.1, and four above 7 [3, 33, 44, 54, 61, 68, 89, 98, 108, 113, 130, 137, 142, 143]. The Tegner score was used in three studies, all reporting 3 as mean or median score at final follow-up [52, 79, 116]. Long et al. [79] also reported a score of 3.5 ± 1.1 eight years postoperatively. The Lower Extremity Activity Scale (LEAS) was used in four studies: three reported results varying between 10.9 and 13.7, which equals a score of 5 or 6 on the UCLA scale [44, 58, 101], and Dubin et al. [129] reported a score of 8.7 in subjects with a preoperative LEAS <10 and 10.8 in those with a preoperative LEAS ≥10.

## Sports after total knee arthroplasty

Sports participation rate was reported by five articles and varied between 70% and 85% [52, 54, 104, 116, 137]. Hepperger et al. [52] stated that 83% of subjects participated in sports occasionally and 70% at least twice a week. The remaining four articles were not clear about their definition of sports participation. In two articles regular exercise was considered as participating in sports, and two articles provided no information at all about their definition. Regarding intensity of the activities, Ristolainen et al. [104] reported that 68% of subjects played sports at walking intensity and 14% at high intensity.

Time spent on sports was reported in two articles: Pioger et al. [98] reported 10.2 ± 6.6 hours/week in a population of active golfers, Mayr et al. [88] reported a mean time spent on sports participation of 5.3 hours/week. Number of sessions per week in the population of Mayr et al. [88] was 3.5, while Ristolainen et al. [104] reported that over half of their subjects participated in sports more than 10 times per month. Three other articles reported on number of sessions per week: 2–3 and 4–6 sessions was the most frequent choice in the studies of Hepperger et al. [52] and Jassim et al. [61], while 1–2 sessions for low-intensity and moderate-intensity activities and 3–4 sessions for strenuous activities was the most frequent choice by the subjects of Smith et al. [18]. Mean session duration was <1 hour for half of the subjects and >1 hour for the other half, as reported by Ristolainen et al. [104].

Type of sports performed was reported by seven articles [28, 33, 52, 54, 65, 88, 116]. Low-impact activities were performed most often, with high participation rates for cycling, swimming and walking, while some studies also found high participations rates for fitness, gymnastics and aqua aerobics. For medium-impact sports, high variation of participation in hiking was found, ranging from 3.6% to 70%. Dancing had a participation rate >25% in two of the seven articles reporting, while downhill and cross-country skiing was performed by >10% of subjects in three out of seven articles. High-impact sports participation rates were reported to be <5% except by Mayr et al. [88], with racket sports performed by 20% of subjects.

## Physical activity after unilateral knee arthroplasty

Amount of activity using objective measures or questionnaires was not reported in any of the articles. The UCLA score was used in 13 studies, 11 reporting an average between 6 and 7, which means regular participation in moderate activities [31, 39, 54, 59, 60, 70, 73, 95, 97, 119, 120, 135, 141]. Similar results were found in seven articles using the Tegner activity scale [16, 60, 70, 95, 119, 120, 135].

## Sports after unilateral knee arthroplasty

Sports participation rates were reported in 12 articles. Seven of those found rates above 85%, one above 80%, two articles above 70%, and two articles reported a rate of 60% [31, 54, 60, 70, 73, 78, 95, 97, 119, 120, 135, 141]. None of those articles gave a specific definition of sports participation considering frequency or intensity, and only Jahnke et al. [60] provided information

that 87% participated weekly, while total sports participation in that study was 93%. Intensity was given by one study reporting that 100% of subjects participated at a recreational level [78].

No articles reported total participation time per week. Frequency of participation was reported in six articles. Mean number of sessions per week, reported in five articles, varied between 1.9 and 3 sessions/week [70, 78, 119, 135, 141]. Walker et al. [120] reported that 53% subjects had three or more sessions. Regarding length of sessions, two separate studies by Walker et al. [119, 120] reported that 44% and 45% of subjects, respectively, were active for at least 1 hour/session, while Kim et al. [70] reported a mean of 1.3 ± 0.7 hours/session compared to the average 43 and 45.9 min/session reported by Lo Presti et al. and Zimmerer et al. [70, 78, 119, 120, 141]. Two articles reported on frequency and session length per individual sport; those results are shown in S4 Appendix [54, 60].

Type of sport performed was reported in eleven articles [31, 39, 54, 60, 70, 73, 78, 95, 97, 119, 120]. Most subjects participated in low-impact sports, especially cycling, swimming and walking, with some higher rates also seen in fitness. Hiking had the highest participation rates for medium-impact sports, while high-impact sports were performed by less than 10% of subjects, except for jogging as reported by Felts et al. and Jahnke et al. [39, 60].

## Physical activity and sports after mixed arthroplasties

Twelve articles reported on subjects with various types of arthroplasties [22, 27, 34, 53, 61, 66, 80, 91, 99, 102, 105, 111]. Most results found in the studies using more than one type of arthroplasty were, as could be expected, comparable to the results described above. Exception was Robertsen et al., who reported a high number of steps and participation in high-impact sports, but they included active persons [105].

## Discussion

Most of the 125 included articles reported on THA, followed by TKA, with a minority reporting on HRA and UKA. PA levels reported were relatively comparable for THA and TKA, and showed a low-to-moderately active population, mostly performing 5000–7000 steps/day. Time spent in PA was mostly of low intensity. Roughly 50% of subjects met health-enhancing PA guidelines. Participants spent the largest part of their day sedentarily.

Sports participation was relatively high for all arthroplasty types, with most articles reporting rates above 70% although the definitions used in the articles varied. Regarding sports participation, by far most subjects engaged in low-impact sports like walking, cycling and swimming at a recreational level roughly 3 hours/week – about three 1-hour sessions/week.

The general quality of the included studies was fair. Most studies included subjects in consecutive fashion, providing a representative sample. However, sample sizes and response rates were low in about half of the studies, while non-response analysis was only conducted in 33 studies. Quality of measurement tools and reporting was good in most cases.

### Physical activity

PA was relatively similar for subjects after THA and TKA, yet the number of steps seemed to be slightly higher for those after TKA while the UCLA score was slightly higher for those after THA. Participants spent most of their days sedentarily, with very limited time spent in MVPA. Adhering to health-enhancing PA guidelines is important for physical as well as mental health. When comparing the time being active to guidelines, seven out of 11 articles reported that only about 40–50% of the included subjects met the guidelines. Mean time spent active varied between 150 and 1525 min/week, most of it performed at a low intensity. The step count of subjects after THA and TKA ranged between 5000 and 7000 steps/day in both groups.

According to Tudor-Locke et al., 5000–7499 steps/day is "typical of daily activity excluding sports/exercise and might be considered low active" [144]. This is consistent with the finding that the majority of the steps taken and activities performed are of low intensity, with only a very small part performed at vigorous intensity. Also, those studies using one-item questionnaires commonly reported that subjects sometimes or regularly participated in moderate activities like swimming, housework, and shopping after THA, TKA, and UKA, which implies participation in activities of daily living. Persons after HRA indicated regular participation in active events, yet the only studies performing objective measures involving a comparable study group found a relatively inactive population with a mean of only 5546 steps/day [132, 133]. As this study included only 16 persons, results should be interpreted with caution. In general, it can be concluded that persons after THA or TKA are low-active, performing activities of daily living but without performing activities at higher intensities.

Sedentary behavior was only reported by ten articles. All found that most persons spent the largest part of the day sedentarily. Increasing attention is directed to the negative effects of sitting for long periods, and the importance of reducing sedentary behavior is now also included in the latest health-enhancing PA guidelines of the World Health Organization [11]. This review, however, shows that only few studies are focused on sedentary behavior in this population. It is of importance to learn more about which persons are at risk for a sedentary lifestyle, and how to lower the time spend sedentarily.

Time spent active was measured in only a few studies, which used a variety of assessment methods and a variety of definitions of PA. The difference in methods was especially apparent when comparing the outcomes of objective and self-reported assessment tools of time persons spent active, as the range varied from about 150 to 1500 min/week, with self-reported assessment tools generally yielding higher outcomes. The different measurement methods also seemed to influence the number of persons meeting health-enhancing PA guidelines. Articles using self-reported tools all found about 40–50% of subjects to meet the guidelines, while four out of five studies using objective tools found a much smaller proportion. This trend was seen for both THA and TKA. Although it is plausible that these objective studies slightly underestimated the activity of their participants – as not all activities, like swimming and bicycling, might be correctly registered – it is also known that people tend to overestimate their activity when using self-reported measurement methods [145, 146]. Results showing 50% of subjects meeting guidelines might therefore be too optimistic. With this in mind, it can be concluded that the number of persons after THA or TKA meeting health-enhancing PA guidelines is lower than the reported 61% for the European adult population [12]. Considering the measurement methods used, it can also be questioned what one-item questionnaires providing a general indication of PA tell us. Although the UCLA score provides a general indication of PA, studies have shown that it has only a weak correlation with the number of strides per day [75, 147]. Despite these kinds of questionnaires having the advantage of being used in large measurement groups, they lack specific information about PA and sedentary behavior and are subject to response bias. It should thus be taken into account that the character and quality of the measurement tools used could have had a large impact on the results reported.

## Sports activity

Sports participation rates were high for all arthroplasty types, with most articles reporting rates above 70%. Participation rates after THA and TKA seemed comparable, while persons after HRA and UKA often reported higher participation levels above 80 or even 90%. That in general more than 70% of subjects were reported to participate in sports is surprising, as in Europe on average 47% of the adult population and only 28% of the population 55 and older state they

participate in any sport at least once a week [148]. The unclear and variable definition of sports participation used probably caused this discrepancy. For example, most of the articles also included activities like walking and bicycling as sports activities, which can also be performed to commute. However, the reason they were performed (e.g. as commuting or as sporting activity) and the intensity of the activity were not always reported. Secondly, many articles did not specify how regularly subjects had to participate to be considered as participating in sports. Future research on this topic should clearly state what definition of sports participation is used in type of sport included, frequency and intensity.

Considering the type of sport performed, most was recreational and light-impact. Also, some medium-impact sports were reported relatively often, especially hiking. Participation in hiking, as well as skiing, was especially high in studies that included populations in mountainous regions. This is not surprising, as cultural and geographical factors play a role in the type of sports people engage in. For high-impact sports, low participation rates were found which seldom exceeded 10%. Only in studies that included highly active individuals or those participating in high-impact sports preoperatively, were higher participation levels in high-impact sports found postoperatively. Although most of these sports might not be generally recommended by most surgeons, preoperative experience with a specific sport is a commonly considered factor when weighing whether someone is advised to participate in that sport or not [149, 150]. Individuals do seem to follow this advice.

## Strengths, limitations and future research

This review presents an overview of the current knowledge about the PA performed by persons after THA or TKA. Due to the fact that we extracted the data about PA at one moment in time, with a large variation in follow-up time, and the large variety in outcome measures used in the studies, it was not possible to perform a meta-analysis. Despite the broad search strategy, one article including HRA and no articles including UKA were found considering activity level in terms of time spent. It is important for more information to be provided about the activity of these groups in the future. This especially applies to UKA, as negative outcomes after HRA have resulted in a ban of HRA in several countries, including the Netherlands [151]. Relatively few studies focused on more detailed aspects of PA and sports, like time spent and intensity of activities, using a variety of assessment methods. Future research might aim to extend the knowledge on this topic, using clear definitions of PA and sports participation and more detailed information about time, frequency and intensity. Attention should likewise focus on the proportion of persons meeting health-enhancing PA guidelines, including time spent sedentarily, given the importance of meeting guidelines for the general health and fitness of this population.

By having a minimum follow-up time we aimed to find less unwanted variety because of improvements in recovery in the first year of rehabilitation. Still, the included articles have shown a large variety in study populations, which makes it harder to synthesize results about activity performed by persons after THA or TKA. And yet this large variety in populations has increased the generalizability of the findings, so results might fit the overwhelming majority of persons after THA or TKA. It is also important to keep in mind that every person is different and may therefore have individual needs, facilitators and barriers when it comes to activity.

## Conclusion

General activity levels of persons after THA or TKA seem to be low, performing activities of daily living but without performing activities at higher intensities. Less than half of subjects seemed to perform the advised amount of PA according to health-enhancing PA guidelines.

The majority of the PA performed was of low intensity, while most of the day was spent sedentarily. Consequently, persons should be stimulated to limit their time being sedentary, in line with the latest WHO guidelines [11]. Still, sports participation levels were high. Many articles were unclear about the definition of sports participation, which could have led to overestimation. Time spent in sports was about 3 hours/week, divided over three 1-hour sessions. Low-impact activities at a recreational level like walking and cycling were favored. The low PA levels found show that there is room for improvement to stimulate persons to become physically active following hip or knee arthroplasty. Persons have to strive to comply with the WHO guidelines [11] with respect to the amount and intensity of physical activity taking into account that excessive or inappropriate PA can negatively influence prosthetic wear and loosening, affecting the longevity of the hip or knee prosthesis [152–154].

## Supporting information

**S1 Appendix. Search strategy.**
(PDF)

**S2 Appendix. Methodological quality tool.**
(PDF)

**S3 Appendix. Quality assessment.**
(PDF)

**S4 Appendix. Overview of characteristics and results of studies reporting on amount and type of physical activity.**
(PDF)

**S5 Appendix. Results on amount and intensity of physical activity.**
(PDF)

**S6 Appendix. Overview of one-item questionnaires giving a general indication of overall physical activity performed.**
(PDF)

**S7 Appendix. Sports activity categorized into levels of sports participation, time spent playing sports (including session length and frequency), and impact and intensity.**
(PDF)

**S8 Appendix. Participation rates per individual sport.**
(PDF)

**S9 Appendix. Prisma checklist.**
(PDF)

## Acknowledgments

**PAIR study group**:

Giuseppe Barone, Department for Life Quality Studies, University of Bologna, Campus of Rimini, Rimini, Italy.

Francesco Benvenuti, Medea, Florence, Italy.

Mihai Berteanu, Carol Davila University of Medicine and Pharmacy, Bucharest, Romania.

Laura Bragonzoni, Lead author (laura.bragonzoni4@unibo.it), Department for Life Quality Studies, University of Bologna, Campus of Rimini, Rimini, Italy.

Ileana Ciobanu, Carol Davila University of Medicine and Pharmacy, Bucharest, Romania.

Dante Dallari, Rizzoli Orthopaedic Institute, Bologna, Italy.

Ani Dimitrova, Know and Can Association, Bulgaria.

Ivo Dimitrov, Know and Can Association, Bulgaria.

Simona Geli, Medea, Florence, Italy.

Jorunn Lægdheim. Helbostad, Norwegian University of Science and Technology, Trondheim, Norway.

Alina Iliescu, Carol Davila University of Medicine and Pharmacy, Bucharest, Romania.

Pasqualino Maietta Latessa, Department for Life Quality Studies, University of Bologna, Campus of Rimini, Rimini, Italy.

Andreea Marin, Carol Davila University of Medicine and Pharmacy, Bucharest, Romania.

Alessandro Mazzotta, Rizzoli Orthopaedic Institute, Bologna, Italy.

Ann-Katrin Stensdotter, Norwegian University of Science and Technology, Trondheim, Norway.

Odd Magne Hals, Norwegian University of Science and Technology, Trondheim, Norway.

Håvard Østerås, Norwegian University of Science and Technology, Trondheim, Norway.

Cristiano Paggetti, Medea, Florence, Italy.

Erika Pinelli, Department for Life Quality Studies, University of Bologna, Campus of Rimini, Rimini, Italy.

Nataliya Shalamanova, Know and Can Association, Bulgaria.

Rumyana Shalamanova, Know and Can Association, Bulgaria.

Claudio Stefanelli, Department for Life Quality Studies, University of Bologna, Campus of Rimini, Rimini, Italy.

Matei Teodorescu, Carol Davila University of Medicine and Pharmacy, Bucharest, Romania.

Nikolay Todorov, Know and Can Association, Bulgaria.

Stefania Toselli, Department for Life Quality Studies, University of Bologna, Campus of Rimini, Rimini, Italy; Department of Biomedical and Neuromotor Science, University of Bologna, Bologna, Italy.

Maya Tsvetanova, Know and Can Association, Bulgaria.

Monica Unsgaard-Tøndel, Norwegian University of Science and Technology, Trondheim, Norway.

Lora Yoncheva, Know and Can Association, Bulgaria.

Raffaele Zinno. Department for Life Quality Studies, University of Bologna, Campus of Rimini, Rimini, Italy.

## Author Contributions

**Conceptualization:** Yvet Mooiweer, Inge van den Akker-Scheek, Martin Stevens.

**Data curation:** Yvet Mooiweer.

**Formal analysis:** Yvet Mooiweer.

**Funding acquisition:** Martin Stevens.

**Investigation:** Yvet Mooiweer.

**Methodology:** Yvet Mooiweer, Martin Stevens.

**Project administration:** Martin Stevens.

**Supervision:** Inge van den Akker-Scheek, Martin Stevens.

**Writing – original draft:** Yvet Mooiweer.

**Writing – review & editing:** Yvet Mooiweer, Inge van den Akker-Scheek, Martin Stevens.

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
