## [Decision Letter · Decision Letter 0]

19 Oct 2021

PONE-D-21-28882Amount and type of physical activity and sports performed by persons after hip or knee arthroplasty − a systematic reviewPLOS ONE

Dear Dr. Stevens,

Thank you for submitting your manuscript to PLOS ONE. After careful consideration, we feel that it has merit but does not fully meet PLOS ONE’s publication criteria as it currently stands. Therefore, we invite you to submit a revised version of the manuscript that addresses the points raised during the review process.

We look forward to receiving your revised manuscript.

Kind regards,

Sinan Kardeş, M.D.

Academic Editor

PLOS ONE

Journal Requirements:

4. One of the noted authors is a group or consortium “PAIR study group”. In addition to naming the author group, please list the individual authors and affiliations within this group in the acknowledgments section of your manuscript. Please also indicate clearly a lead author for this group along with a contact email address.

Reviewers' comments:

Reviewer's Responses to Questions

**Comments to the Author**

1. Is the manuscript technically sound, and do the data support the conclusions?

Reviewer #1: Partly

Reviewer #2: Yes

2. Has the statistical analysis been performed appropriately and rigorously? 

Reviewer #1: No

Reviewer #2: N/A

3. Have the authors made all data underlying the findings in their manuscript fully available?

Reviewer #1: Yes

Reviewer #2: Yes

4. Is the manuscript presented in an intelligible fashion and written in standard English?

Reviewer #1: Yes

Reviewer #2: Yes

5. Review Comments to the Author

Reviewer #1: Amount and type of physical activity and sports performed by persons after hip or knee

arthroplasty − a systematic review

The manuscript is an interesting systematic review (narrative) regarding the physical activity levels, sedentary behaviour, and sport participation of people with Total hip arthroplasty (THA) and total knee arthroplasty (TKA) after one-year post-surgery. The title, objectives and conclusions are well aligned, but there might be room for improvement in the results section. The title is very similar to previous studies published; however, the authors highlighted the novelty. The methods considered an unbiased sampling of existing literature. Statistical analysis was not detailed or not performed. Authors appear to meet research integrity. Congratulations on their effort to summarise a huge quantity of papers.

Please find below suggestions that might improve the manuscript and make it more enjoyable for readers:

Major issues:

• Please clarify whether an inclusion criterion was physical activity measurement performed after one-year post-surgery onwards. I think this is the essence of the study and the main difference in respect to other published studies. The authors might consider reflecting the manuscript’s novelty in the title.

• Please discuss any changes or amendments to the protocol published in Prospero, especially regarding the initially proposed authors.

• Please include a paragraph regarding the statistical analysis or data synthesis

• The results’ section is about 11 pages long, which is considerably extensive. In my opinion, the authors failed to synthesise the data.

Minor issues:

Introduction

In the first line of the introduction, the authors wrote that both, Total hip arthroplasty (THA) and total knee arthroplasty (TKA) are cost-effective, however, the references provided were not focused on cost-effectiveness analysis. Similarly, the same references do not endorse the statement “Total hip arthroplasty (THA) and total knee arthroplasty (TKA) improve the ability to stay physically active”.

Methods

• Please state the criteria of the type of study design included in the review.

• The data extraction section was brilliantly written.

Results

• Please check a possible typo in line 160 “(98, 80%)”.

• Please expand about “found comparable results” in line 221.

• Please explain, in the methods section, about the “non-response analysis” mentioned in line 156.

• The tables summarise valuable information, congratulations to the authors for their effort.

• Please display the first header row on each page to ease reading the tables

• Please explain how to interpret the outcome in the appendixes, especially number 6 (eg question used to assess PA, min. value, max value, range, higher score representing more (and more intense) PA)

Discussion

It shows a better summary of the review.

• Please include a statement about the decision not to perform a meta-analysis. It might be a limitation,

• What are the clinical or practical repercussions/recommendations regarding the findings of low intensity and low activity levels for those populations?

Conclusions

• How was calculated the result for: “Time spent in sports was about 3 hours/week” in line 527?

• Can the authors conclude or give an idea of what percentage of these populations or papers meet the WHO physical activity guidelines?

Reviewer #2: The authors have assessed the quality of the articles using the adapted tool from Borghouts et al. This tool was developed in 1998. Why did the authors select this tool? I think that a more up-to-date tool could have been used instead of this tool.

6. PLOS authors have the option to publish the peer review history of their article (what does this mean?). If published, this will include your full peer review and any attached files.

Reviewer #1: **Yes: **Carlos Mesa Castrillon

Reviewer #2: No

---

## [Author Response · Author response to Decision Letter 0]

24 Nov 2021

Reviewer #1: Amount and type of physical activity and sports performed by persons after hip or knee arthroplasty − a systematic review

The manuscript is an interesting systematic review (narrative) regarding the physical activity levels, sedentary behaviour, and sport participation of people with Total hip arthroplasty (THA) and total knee arthroplasty (TKA) after one-year post-surgery. The title, objectives and conclusions are well aligned, but there might be room for improvement in the results section. The title is very similar to previous studies published; however, the authors highlighted the novelty. The methods considered an unbiased sampling of existing literature. Statistical analysis was not detailed or not performed. Authors appear to meet research integrity. Congratulations on their effort to summarise a huge quantity of papers.

Please find below suggestions that might improve the manuscript and make it more enjoyable for readers:

Major issues:

• Please clarify whether an inclusion criterion was physical activity measurement performed after one-year post-surgery onwards. I think this is the essence of the study and the main difference in respect to other published studies. The authors might consider reflecting the manuscript’s novelty in the title.

Thanks for the suggestion. We changed the title as suggested by the reviewer. In addition we included a statement at the end of the introduction section in which we formulate the objective of the study “from one year forward after THA or TKA” 

The reviewer is right. As mentioned in Prospero we included studies in which patients had finished the rehabilitation after hip or knee arthroplasty, this was defined as 1 year postoperative. In general one year postoperative is the point in time that in general patients are fully recovered.

• Please discuss any changes or amendments to the protocol published in Prospero, especially regarding the initially proposed authors.

The reviewer is right in Prospero a different title is submitted. “State of knowledge regarding physical activities after rehabilitation in patients after hip or total knee arthroplasty, a systematic review”. Consequently four research questions were formulated. In the end we decided to execute two separate systematic reviews all covering a different aspect under this umbrella. The current systematic review is one of the two, with the focus on the amount and type of physical activity and sports. The other focuses on recommendations with respect to becoming active in physical activity and sports.

With respect to the authors mentioned the reviewer is also right. This has also to do with the subdivision in two separate reviews. Roosmarijn Geerlings is not an author on the current paper but is on the other paper. Which is currently under review with another journal. 

Finally to assess the methodological quality an adapted version of the tool developed by Borghouts et al. 1998 was used instead of the proposed tools in Prospero. In the end we choose for the tool of Borghouts as it can cover a broad range of research designs. 

In the meantime we also updated our Prospero registration.

• Please include a paragraph regarding the statistical analysis or data synthesis

Due to the heterogeneity of the data, patient groups measured, and measurement tools used a synthesis of the data was not possible, and statistical analyses were consequently not applicable. As a result the current review provides a narrative synthesis of the data. Consequently we deemed it not necessary to include a separate paragraph with respect to statistical analysis or data synthesis.

• The results’ section is about 11 pages long, which is considerably extensive. In my opinion, the authors failed to synthesise the data.

See also our response to the former remark. In the end we performed a narrative review. In fact we tried to do this narrative review as concise as possible. However due to the subdivision in physical activity and sports and subsequently in the different types of arthroplasty it still is indeed quite long. On the other hand due to this subdivision in separate parts, the reader can easily find the relevant information for a certain type of arthroplasty and respectively physical activity and sports.

Minor issues:

Introduction

In the first line of the introduction, the authors wrote that both, Total hip arthroplasty (THA) and total knee arthroplasty (TKA) are cost-effective, however, the references provided were not focused on cost-effectiveness analysis. Similarly, the same references do not endorse the statement “Total hip arthroplasty (THA) and total knee arthroplasty (TKA) improve the ability to stay physically active”.

In accordance with the suggestion of the reviewer we updated the references. With respect to cost-effectiveness: 

Kamaruzaman H, Kinghorn P, Oppong R. Cost-effectiveness of surgical interventions for the management of osteoarthritis: a systematic review of the literature. BMC Musculoskelet Disord 2017 181. 2017 May 10;18(1):1–17.

With respect to the ability to stay physically active:

Stevens M, Reininga IH, Bulstra SK, Wagenmakers R, van den Akker-Scheek I. Physical activity participation among patients after total hip and knee arthroplasty. Clin Geriatr Med. 2012 Aug;28(3):509-20. doi: 10.1016/j.cger.2012.05.003. Epub 2012 May 24.

Methods

• Please state the criteria of the type of study design included in the review.

In principle all kind of study designs were included with the exclusion of review articles, case reports and study protocols, We added a remark in the methods section.

• The data extraction section was brilliantly written.

Thanks for your positive response.

Results

• Please check a possible typo in line 160 “(98, 80%)”.

The reviewer is right this is a mistake, we made a correction.

• Please expand about “found comparable results” in line 221.

It was meant that comparable results were found as in the studies using the UCLA. We adapted the sentence: Nine articles used similar scores (Tegner activity scale, Grimby scale, Lower Extremity Activity Scale (LEAS), Sports Activity Index, Weighted Activity Score) and their results are comparable to those of the studies using the UCLA. 

• Please explain, in the methods section, about the “non-response analysis” mentioned in line 156.

We do not exactly understand what the reviewer intends with this remark. The fact if a non-response analysis was done or not is considered an indication of the methodological quality of the study at hand. As reported this was only the case in a minority (27%). 

• The tables summarise valuable information, congratulations to the authors for their effort.

Thanks for your positive response.

• Please display the first header row on each page to ease reading the tables

This is a good suggestion, we changed the tables accordingly.

• Please explain how to interpret the outcome in the appendixes, especially number 6 (eg question used to assess PA, min. value, max value, range, higher score representing more (and more intense) PA)

As suggested we included the explanations in appendix 6 and also in appendix 4. 

Discussion

It shows a better summary of the review.

Thanks for your positive response.

• Please include a statement about the decision not to perform a meta-analysis. It might be a limitation,

As suggested by the reviewer we included a statement why a meta-analysis was not executed. “Due to the fact that we extracted the data about PA at one moment in time, with a large variation in follow-up time, and the large variety in outcome measures used in the studies, it was not possible to perform a meta-analysis”.

• What are the clinical or practical repercussions/recommendations regarding the findings of low intensity and low activity levels for those populations? 

In the conclusion section of the paper we added two statements in line with the WHO 2020 guidelines: first we added a statement that the amount of time spent being sedentary has to be limited. Secondly we added a statement that because of the low PA levels found in the review persons have to strive to comply with the WHO 2020 guidelines with respect to the amount and intensity of physical activity as recommended. But that they have to take into account that excessive or inappropriate PA can negatively influence prosthetic wear and loosening, affecting the longevity of the hip or knee prosthesis. 

Conclusions

• How was calculated the result for: “Time spent in sports was about 3 hours/week” in line 527?

This was not calculated, however this was based on the fact that this was the case in a majority of the included papers. That’s why we also stated “about 3 hours/week”.

• Can the authors conclude or give an idea of what percentage of these populations or papers meet the WHO physical activity guidelines?

This is a very interesting suggestion however we are not able to give a substantiated answer. This has primarily to do with the fact that the information derived from the included papers is not sufficient enough. Secondly, in fact in only a (very) small amount of the papers information is reported with respect to complying with guidelines, and finally, if reported, these guidelines in the past differed from the most recent, as mentioned in our paper. 

Reviewer #2: The authors have assessed the quality of the articles using the adapted tool from Borghouts et al. This tool was developed in 1998. Why did the authors select this tool? I think that a more up-to-date tool could have been used instead of this tool.

In principle the reviewer has a point. However we could not find a more recent tool that could cover the broad range of research designs that we encountered in our systematic review. So in the end, based on pragmatic reasons we decided to use an adapted version of the tool developed by Borghouts et al. 1998.

---

## [Decision Letter · Decision Letter 1]

10 Dec 2021

Amount and type of physical activity and sports from one year forward after hip or knee arthroplasty − a systematic review

PONE-D-21-28882R1

Dear Dr. Stevens,

We’re pleased to inform you that your manuscript has been judged scientifically suitable for publication and will be formally accepted for publication once it meets all outstanding technical requirements.

Kind regards,

Sinan Kardeş, M.D.

Academic Editor

PLOS ONE

Additional Editor Comments (optional):

Reviewers' comments:

Reviewer's Responses to Questions

**Comments to the Author**

1. If the authors have adequately addressed your comments raised in a previous round of review and you feel that this manuscript is now acceptable for publication, you may indicate that here to bypass the “Comments to the Author” section, enter your conflict of interest statement in the “Confidential to Editor” section, and submit your "Accept" recommendation.

Reviewer #1: All comments have been addressed

Reviewer #2: All comments have been addressed

2. Is the manuscript technically sound, and do the data support the conclusions?

Reviewer #1: Yes

Reviewer #2: Yes

3. Has the statistical analysis been performed appropriately and rigorously? 

Reviewer #1: N/A

Reviewer #2: No

4. Have the authors made all data underlying the findings in their manuscript fully available?

Reviewer #1: Yes

Reviewer #2: Yes

5. Is the manuscript presented in an intelligible fashion and written in standard English?

Reviewer #1: Yes

Reviewer #2: Yes

6. Review Comments to the Author

Reviewer #1: Thanks to the authors that adressed all comments succesfully. They declared the statistically analysis did not apply to the narrative review and that the PROSPERO protocol was updated.

Reviewer #2: The revised manuscript seems adequate scientifical value. I would like to thank the authors for the addmission to the reviewer's suggestions.

7. PLOS authors have the option to publish the peer review history of their article (what does this mean?). If published, this will include your full peer review and any attached files.

Reviewer #1: **Yes: **Carlos Mesa Castrillon

Reviewer #2: No

---

## [Editor Report · Acceptance letter]

16 Dec 2021

PONE-D-21-28882R1 

Amount and type of physical activity and sports from one year forward after hip or knee arthroplasty − a systematic review 

Dear Dr. Stevens:

I'm pleased to inform you that your manuscript has been deemed suitable for publication in PLOS ONE. Congratulations! Your manuscript is now with our production department. 

Kind regards, 

on behalf of

Dr. Sinan Kardeş 

Academic Editor

PLOS ONE